# A SYNAPTIC NEURAL NETWORK AND SYNAPSE LEARNING

## ABSTRACT

A Synaptic Neural Network (SynaNN) consists of synapses and neurons. Inspired by the synapse research of neuroscience, we built a synapse model with a nonlinear synapse function of excitatory and inhibitory channel probabilities. Introduced the concept of surprisal space and constructed a commutative diagram, we proved that the inhibitory probability function -log(1-exp(-x)) in surprisal space is the topologically conjugate function of the inhibitory complementary probability 1-x in probability space. Furthermore, we found that the derivative of the synapse over the parameter in the surprisal space is equal to the negative Bose-Einstein distribution. In addition, we constructed a fully connected synapse graph (tensor) as a synapse block of a synaptic neural network. Moreover, we proved the gradient formula of a cross-entropy loss function over parameters, so synapse learning can work with the gradient descent and backpropagation algorithms. In the proof-of-concept experiment, we performed an MNIST training and testing on the MLP model with synapse network as hidden layers.

## 1 INTRODUCTION

Synapses play an important role in biological neural networks (Hubel & Kandel (1979)). They are joint points of neurons' connection with the capability of learning and memory in neural networks. Based on the analysis of excitatory and inhibitory channels of synapses(Hubel & Kandel (1979)), we proposed a probability model (Feller (2008) for probability introduction) of the synapse together with a non-linear function of excitatory and inhibitory probabilities (Li (2017)(synapse function)). Inspired by the concept of surprisal from (Jones (1979)(self-information), Levine (2009), Bernstein & Levine (1972)(surprisal analysis), Levy (2008)(surprisal theory in language)) or negative logarithmic space (Miyashita et al. (2016)), we proposed the concept of surprisal space and represented the synapse function as the addition of the excitatory function and inhibitory function in the surprisal space. By applying a commutative diagram, we figured out the fine structure of inhibitory function and proved that it was the topologically conjugate function of an inhibitory function. Moreover, we discovered (rediscovered) that the derivative of the inhibitory function over parameter was equal to the negative Bose-Einstein distribution (Nave (2018)). Furthermore, we constructed a fully connected synapse graph and figured out its synapse tensor expression. From synapse tensor and a cross-entropy loss function, we found and proved its gradient formula that was the basis for gradient descent learning and using backpropagation algorithm. In surprisal space, the parameter (weight) updating for learning was the addition of the value of the negative Bose-Einstein distribution. Finally, we designed the program to implement a Multiple Layer Perceptrons (MLP) (Minsky et al. (2017)) for MNIST (LeCun et al. (2010)) and tested it to achieve the near equal accuracy of standard MLP in the same setting.

### 1.1 BACKGROUND

Hodgkin and Huxley presented a physiological neuron model that described the electronic potential of the membrane between a neuron and a synapse with a differential equation (Hodgkin & Huxley (1952)). Later, neuron scientists have found that a synapse might have a complicated channel structure with rich chemical and electronic properties (Lodish et al. (1995)(biological synapse), Destexhe et al. (1994)(computing synaptic conductances), Abbott & Nelson (2000)(synaptic plasticity)). Other synapse models based on differential equations had been proposed and been simulated by analogy

circuits like Spiking Neural Network (SNN) (Kumar (2017) (differential equations), Lin et al. (2018) (Intel's SNN Loihi)). In these approaches, synapses acted as linear amplifiers with adjustable coefficients. An example was the analog circuit implementation of Hopfield neural network (Hopfield & Tank (1986)(analog neural circuits)).

In this paper, we proposed a simple synapse model represented by the joint opening probability of excitatory and inhibitory channels in a synapse. It was described as a non-linear computable synapse function. This neuroscience-inspired model was motivated on our unpublished research to solve optimization problems by neural networks.

To do learning by gradient descent and backpropagation algorithm (Goodfellow et al. (2016)(book on deep learning)), because of the differentiable of the synapse function in the synaptic neural network, we could compute Jacobian matrix explicitly and compute the gradient of the cross-entropy loss function over parameters. Therefore, we provided a detailed proof of the formula of gradients in Appendix A

In the process of analyzing Jacobian matrix, we found that the derivative of the inhibitory function $log(1 - e^{-x})$ was equal to the $1/(e^x - 1)$ which was the formula of Bose-Einstein distribution (Einstein (1924)(quantum ideal gas)). In statistical physics and thermodynamics, Bose-Einstein distribution had been concluded from the geometric series of the Bose statistics.

A dual space analysis was an efficient scientific method. After successful expressing fully-connected synapse network in a logarithmic matrix, we started to consider log function and log space. The concept of surprisal (where was the first definition of surprisal?), which was the measurement of surprise from Information Theory (Shannon (1948)), gave us hints. Original surprisal was defined on the random variable, however, it was convenient to consider the probability itself as a variable. So we introduced the surprisal space with a mapping function -log(p). The motivation was to transform any points from probability space to surprisal space and in reverse.

In surprisal space, a synapse function was the addition of an excitatory identity function and an inhibitory function. Although we had figured out the inhibitory function being $-log(1 - e^{-x})$, we wanted to know its structure and what class it belonged to.

This was a procedure that we rediscovered the way to construct a commutative diagram for synapse inhibitory function Diagram (2.2.3). In 1903, Mathematician Bertrand Russell presented the first commutative diagram in his book (Russell (2009)) before the category theory. You can find a good introduction of applied category theory by (Bradley (2018)). In this paper, we did not require to know category theory.

The basic idea was to given two spaces and two points in source space which have corresponding points in target space by a continuous and inverse mapping function from source space to target space, plus, a function that maps start point to the endpoint in the same source space. Our question is to find the function that maps the corresponding start point to the corresponding endpoint in the target space (refer to diagram 2.2.3). There are two paths from source start point to target endpoint: one is from top-left, go right and down to bottom-right; another is from top-left, go down and right to bottom-right. The solution is to solve the equation that has the same target endpoint.

We found that the synapse inhibitory function $-log(1 - e^{-x})$ was also a topologically conjugate function. Therefore, the synaptic neural network has the same dynamical behavior in both probability space and surprisal space. To convince that the synaptic neural network can work for learning and using the backpropagation algorithm, we proved the gradients of loss function by applying basic calculus. In surprisal space, the negative Bose-Einstein distribution was applied to the updating of parameters in the learning of synaptic neural network. Finally, we implemented a MNIST experiment of MLP to be the proof-of-concept.

## 1.2 CONTRIBUTIONS

1) present a neuroscience-inspired synapse model and a synapse function based on the opening probability of channels. 2) defined surprisal space to link information theory to the synaptic neural network. 3) figure out function $\mathcal{G}(x) = -log(1 - e^{-x})$ as the inhibitory part of a synapse. 4) find the derivative of $\mathcal{G}(x)$ to be the formula of negative Bose-Einstein distribution. 5) discover $\mathcal{G}(x)$ to be a topologically conjugate function of the complementary probability. 6) represent fully-

connected synapse as a synapse tensor. 7) express synapse learning of gradient descent as a negative Bose-Einstein distribution.

## 2 SYNAPTIC NEURAL NETWORK (SYNANN)

A Synaptic Neural Network (SynaNN) contains non-linear synapse networks that connect to neurons. A synapse consists of an input from the excitatory-channel, an input from the inhibitory-channel, and an output channel which sends a value to other synapses or neurons. Synapses may form a graph to receive inputs from neurons and send outputs to other neurons. In advance, many synapse graphs can connect to neurons to construct a neuron graph. In traditional neural network, its synapse graph is simply the wight matrix or tensor.

### 2.1 SYNAPSE IN PROBABILITY SPACE

#### 2.1.1 BIOLOGICAL SYNAPSE

Changes in neurons and synaptic membranes (i.e. potential gate control channel and chemical gate control channel show selectivity and threshold) explain the interactions between neurons and synapses (Torre & Poggio (1978)). The process of the chemical tokens (neurotransmitters) affecting the control channel of the chemical gate is accomplished by a random process of mixing tokens of the small bulbs on the membrane. Because of the randomness, a probabilistic model does make sense for the computational model of the biological synapse (Hubel & Kandel (1979)).

In a synapse, the Na+ channel illustrates the effect of an excitatory-channel. The Na+ channels allow the Na+ ions flow in the membrane and make the conductivity increase, then produce excitatory post-synapse potential. The K+ channels illustrate the effects of inhibitory channels. The K+ channel that lets the K+ ions flow out of the membrane shows the inhibition. This makes the control channel of potential gate closing and generates inhibitory post-potential of the synapse. Other kinds of channels (i.e. Ca channel) have more complicated effects. Biological experiments show that there are only two types of channels in a synapse while a neuron may have more types of channels on the membrane. Experiments illustrate that while a neuron is firing, it generates a series of spiking pulse where the spiking rate (frequency) reflects the strength of stimulation.

#### 2.1.2 NEUROSCIENCE-INSPIRED SYNAPSE MODEL

From neuroscience, there are many types of chemical channels in the membrane of a synapse. They have the following properties: 1) the opening properties of ion channels reflect the activity of synapses. 2) the excitatory and inhibitory channels are two key types of ion channels. 3) the random properties of channels release the statistical behavior of synapses.

From the basic properties of synapses, we proposed the synapse model below:

1) The open probability x of the excitatory channel ($\alpha$-channel) is equal to the number of open excitatory channels divided by the total number of excitatory channels of a synapse. 2) The open probability y of the inhibitory channel ($\beta$-channel) is equal to the number of open inhibitory channels divided by the total number of inhibitory channels of a synapse. 3) The joint probability of a synapse that affects the activation of the connected output neuron is the product of the probability of excitatory channel and the complementary probability of the inhibitory channel. 4) There are two parameters to control excitatory channel and inhibitory channel respectively.

Given two random variables $(X, Y)$, their probabilities $(x, y)$, and two parameters $(\alpha, \beta)$, the joint probability distribution function $S(x, y; \alpha, \beta)$ for X, Y (the joint probability of a synapse that activates the connected neuron) is defined as

$$S(x, y; \alpha, \beta) = \alpha x(1 - \beta y) \tag{1}$$

where $x \in (0, 1)$ is the open probability of all excitatory channels and $\alpha > 0$ is the parameter of the excitatory channels; $y \in (0, 1)$ is the open probability of all inhibitory channels and $\beta \in (0, 1)$ is the parameter of the inhibitory channels. The symbol semicolon ";" separates the variables and parameters in the definition of function S.

## 2.2 SYNAPSE IN SURPRISAL SPACE

Surprisal (self-information) is a measure of the surprise in the unit of bit, nat, or hartley when a random variable is sampled. Surprisal is a fundamental concept of information theory and other basic concepts such as entropy can be represented as the function of surprisal. The concept of surprisal has been successfully used in molecular chemistry and natural language research.

### 2.2.1 SURPRISAL SPACE

Given a random variable X with value x, the probability of occurrence of x is p(x). The standard definitions of **Surprisal** $\mathcal{I}_p(x)$ is the measure of the surprise in the unit of a bit (base 2), a nat (base e), or a hartley (base 10) when the random variable X is sampled at x. Surprisal is the negative logarithmic probability of x such that $\mathcal{I}_p(x) = -log(p(x))$. Ignored random variable X, we can consider p(x) as a variable in **Probability Range Space** or simply called **Probability Space** in the context of this paper which is the open interval (0,1) of real numbers.

**Surprisal Function** is defined as $\mathcal{I} : (0,1) \rightarrow (0,\infty)$ and $\mathcal{I}(x) = -log(x)$ where $x \in (0,1)$ is an open interval in $\mathbb{R}^+$. Its inverse function is $\mathcal{I}^{-1}(u) = e^{-u}$ where $u \in \mathbb{R}^+$. Since surprisal function $\mathcal{I}(x)$ is bijective, exists inverse and is continuous, $\mathcal{I}(x)$ is a **homeomorphism**.

**Surprisal Space** $\mathbb{S}$ is the mapping space of the **Probability Space** $\mathbb{P}$ with the negative logarithmic function which is a bijective mapping from the open interval $(0,1)$ of real numbers to the real open interval $(0,\infty) = R^+$.

$$\mathbb{S} = \{s \in (0,\infty) : s = -log(p), for all\ p \in (0,1)\} \tag{2}$$

The probability space $\mathbb{P}$ and the surprisal space $\mathbb{S}$ are topological spaces of real open interval (0,1) and positive real numbers $\mathbb{R}^+$ that inherit the topology of real line respectively.

### 2.2.2 SURPRISAL SYNAPSE

Given variables $u, v \in \mathbb{S}$ and parameters $\theta, \gamma \in \mathbb{S}$ which are equal to variables $-log(x), -log(y)$ and parameters $-log(\alpha), -log(\beta)$ respectively. The **Surprisal Synapse** $\mathcal{LS}(u, v; \theta, \gamma) \in \mathbb{S}$ is defined as,

$$\mathcal{LS}(u, v; \theta, \gamma) = -log(\mathcal{S}(x, y; \alpha, \beta)) \tag{3}$$

Expanding the right side, there is $\mathcal{LS}(u, v; \theta, \gamma) = (-log(\alpha x)) + (-log(1 - \beta y))$. The first part is an identity mapping plus a parameter. To understand the second part more, we need to figure out its structure and class.

### 2.2.3 TOPOLOGICAL CONJUGACY

**Theorem 1** (Topologically conjugate function). *Given $y = \mathcal{F}(x)$ where $\mathcal{F}(x) = 1 - x$; $x, y \in \mathbb{P}$, $(u, v) = \mathcal{I}(x, y)$ where $u, v \in \mathbb{S}$, and the homeomorphism $\mathcal{I}(x) = -log(x)$ from $\mathbb{P}$ to $\mathbb{S}$, then function $\mathcal{G}$ such that $v = \mathcal{G}(u)$ is*

$$\mathcal{G}(u) = \mathcal{I} \circ \mathcal{F} \circ \mathcal{I}^{-1}(u) = -log(1 - e^{-u}) \tag{4}$$

*Proof.* Building a commutative diagram with the homeomorphism $\mathcal{I}(x)$ below,

$$
\begin{array}{ccc}
x \in \mathbb{P} & \xrightarrow{\mathcal{F}} & y \in \mathbb{P} \\
\mathcal{I} \downarrow & & \downarrow \mathcal{I} \\
u \in \mathbb{S} & \xrightarrow{\mathcal{G}} & v \in \mathbb{S}
\end{array}
$$

The proof is to figure out the equivalent of two paths from x to v. One path is from top x, go right to y and go down to bottom so $v = \mathcal{I}(\mathcal{F}(x))$. Another path is from top x, go down to u and go right to bottom so $v = \mathcal{G} \circ \mathcal{I}$, thus, $\mathcal{I}(\mathcal{F}(x)) = \mathcal{G}(\mathcal{I}(x))$. Let $\circ$ be the composition of functions, the previous

equation is $\mathcal{I} \circ \mathcal{F} = \mathcal{G} \circ \mathcal{I}$. Applying $\mathcal{I}^{-1}$ on both right sides and compute $\mathcal{G}$ on given functions, we proved Eq.(4). □

Given two topological spaces $\mathbb{P}$ and $\mathbb{S}$, continuous function $\mathcal{F} : \mathbb{P} \to \mathbb{P}$ and $\mathcal{G} : \mathbb{S} \to \mathbb{S}$ as well as homeomorphism $\mathcal{I} : \mathbb{P} \to \mathbb{S}$, if $\mathcal{I} \circ \mathcal{F} = \mathcal{G} \circ \mathcal{I}$, then $\mathcal{G}$ is called the topologically conjugated function of the function $\mathcal{F}$ in the standard definition. From Theorem 1, specially $\mathcal{G}(u) = -log(1 - e^{-u})$ is the topologically conjugate function of the complementary probability function $1 - x$. Features:

i) The iterated function $\mathcal{F}$ and its topologically conjugate function $\mathcal{G}$ have the same dynamics. ii) They have the same mapped fixed point where $\mathcal{F} : x = 1/2$ and $\mathcal{G} : u = -log(1/2)$. iii) $\mathcal{I}(x) = -log(x)$ is a infinite differentiable and continuous function in real open interval (0,1).

Let parametric function be $D(u; \theta) = u + \theta$, the surprisal synapse is

$$\mathcal{LS}(u, v; \theta, \gamma) = D(u; \theta) + (\mathcal{I} \circ \mathcal{F} \circ \mathcal{I}^{-1})(D(v; \gamma)) \tag{5}$$

From Eq.(5), the universal function of a surprisal synapse is the addition of the excitatory function and the topologically conjugate inhibitory function in surprisal space. By constructed a commutative diagram, we figured out the elegant structure and topological conjugacy of the function $-log(1-e^{-u})$, which is a new example of the commutative diagram and the topological conjugate function from synaptic neural network. A bridge has been built to connect the synaptic neural network to the category theory, the topology, and the dynamical system.

### 2.2.4 BOSE-EINSTEIN DISTRIBUTION

It is interesting to find the connection between the surprisal synapse and the topologically conjugate function. Furthermore, we are going to figure out the connection between the surprisal synapse and the Bose-Einstein distribution. The Bose-Einstein distribution ($\mathcal{BED}$) is represented as the formula $f(E) = \frac{1}{Ae^{E/kT}-1}$ where f(E) is the probability that a particle has the energy E in temperature T. k is Boltzmann constant, A is the coefficient (Nave (2018)).

**Theorem 2.** *The $\mathcal{BED}$ function is defined as $\mathcal{BED}(v; \gamma) = \frac{1}{e^{\gamma+v}-1}$ where variable $v \in \mathbb{S}$, parameter $\gamma \in \mathbb{S}$, and $v + \gamma \geq ln(2)$, so that $0 \leq \mathcal{BED}(v; \gamma) \leq 1$, then there is $\frac{\partial}{\partial \gamma} \mathcal{G}(D(v; \gamma)) = \mathcal{BED}(v; \gamma)$ or*

$$\frac{\partial}{\partial \gamma}(-log(1 - e^{-D(v;\gamma)})) = \frac{-1}{e^{D(v;\gamma)} - 1} \tag{6}$$

*Proof.* Proved by computing the derivative of the function on left side. □

Recall that $D(v; \gamma) = v + \gamma$, the derivative of the topologically conjugate function $\mathcal{G}$ over parameter $\gamma$ is equal to the negative Bose-Einstein distribution. The gradient of the surprisal synapse $\mathcal{LS}(u, v; \theta, \gamma)$ is

$$(\frac{\partial}{\partial u}; \frac{\partial}{\partial \theta}, \frac{\partial}{\partial v}; \frac{\partial}{\partial \gamma})\mathcal{LS}(u, v; \theta, \gamma) = (1; 1, -\mathcal{BED}(v; \gamma); -\mathcal{BED}(v, \gamma)) \tag{7}$$

This is a connection between surprisal synapse and statistical physics. In physics, $\mathcal{BED}(v; \gamma)$ can be thought of as the probability that boson particles remain in $v$ energy level with an initial value $\gamma$.

### 2.3 SYNAPSE GRAPH AND TENSOR

Generally, a biological neuron consists of a soma, an axon, and dendrites. Synapses are distributed on dendritic trees and the axon connects to other neurons in the longer distance. A ***synapse graph*** is the set of synapses on dendritic trees of a neuron. A synapse can connect its output to an input of a neuron or to an input of another synapse. A synapse has two inputs: one is excitatory input and another is inhibitory input. Typically neurons receive signals via the synapses on dendrites and send out spiking plus to an axon (Hubel & Kandel (1979)).

Assume that the total number of input of the **synapse graph** equals the total number of outputs, the **fully-connected synapse graph** is defined as

$$y_i(\mathbf{x}; \boldsymbol{\beta}_i) = x_i \prod_{j=1}^{n}(1 - \beta_{ij}x_j), \ for \ all \ i \in [1, n] \tag{8}$$

where $\mathbf{x} = (x_1, \cdots, x_n)$, $x_i \in (0, 1)$ and $\mathbf{y} = (y_1, \cdots, y_n)$ are row vectors of probability distribution; $\boldsymbol{\beta}_i = (\beta_{i1}, \cdots, \beta_{in})$, $0 < \beta_{ij} < 1$ are row vectors of parameters; $\boldsymbol{\beta} = matrix\{\beta_{ij}\}$ is the matrix of all parameters. $\boldsymbol{\alpha} = 1$ is assigned to Eq.1 to simplify the computing.

An output $y_i$ of the fully-connected synapse graph is constructed by linking the output of a synapse to the excitatory input of another synapse in a chain while the inhibitory input of each synapse is the output of neuron $x_i$ in series. In the case of the diagonal value $\beta_{ii}$ is zero, there is no self-correlated factor in the $i$th item.

This fully-connected synapse graph represents that only neuron itself acts as excitation all its connected synapses act as inhibition. This follows the observation of neuroscience that most synapses act as inhibition.

**Theorem 3** (Synapse tensor formula). *The following **synapse tensor formula** Eq.9 is equivalent to **fully-connected synapse graph** defined in the Eq.8*

$$log(\boldsymbol{y}) = log(\boldsymbol{x}) + \boldsymbol{1}_{|x|} * log(\boldsymbol{1}_{|\beta|} - diag(\boldsymbol{x}) * \boldsymbol{\beta}^T) \tag{9}$$

*or $\mathcal{I}(\boldsymbol{y}) = \mathcal{I}(\boldsymbol{x}) + \boldsymbol{1}_{|x|} * \mathcal{I}(\boldsymbol{1}_{|\beta|} - diag(\boldsymbol{x}) * \boldsymbol{\beta}^T)$ where $\boldsymbol{x}$, $\boldsymbol{y}$, and $\boldsymbol{\beta}$ are distribution vectors and parameter matrix. $\boldsymbol{\beta}^T$ is the transpose of the matrix $\boldsymbol{\beta}$. $\boldsymbol{1}_{|x|}$ is the row vector of all real number ones and $\boldsymbol{1}_{|\beta|}$ is the matrix of all real number ones that have the same size and dimension of $\boldsymbol{x}$ and $\boldsymbol{\beta}$ respectively. Moreover, the * is the matrix multiplication, diag($\boldsymbol{x}$) is the diagonal matrix of the row vector $\boldsymbol{x}$, and the $log$ is the logarithm of the tensor (matrix).*

*Proof.* Applying the log on both sides of the definition Eq.(8) and completes the matrix multiplications in the ***fully-connected synapse graph***, we proved the formula Eq.(9). Furthermore, by the definition of $\mathcal{I}(x)$, we have the expression of surprisal synapse. □

## 2.4 SYNAPSE LEARNING

To prove that ***synapse learning*** of synaptic neural network is compatible with the standard backpropagation algorithm, we are going to apply cross-entropy as the loss function and use gradient descent to minimize that loss function.

### 2.4.1 GRADIENT OF LOSS FUNCTION

The basic idea of deep learning is to apply gradient descent optimization algorithm to update the parameters of the deep neural network and achieve a global minimization of the loss function (Goodfellow et al. (2016)).

**Theorem 4** (Gradient equation). *Let the loss function $L(\hat{\boldsymbol{o}}, \boldsymbol{o})$ of the **fully-connected synapse graph** Eq.8 be equal to the sum of cross-entropy $L(\hat{\boldsymbol{o}}, \boldsymbol{o}) = -\sum_i \hat{o}_i log(o_i)$, then its parameter gradient is*

$$\frac{\partial L(\hat{\boldsymbol{o}}, \boldsymbol{o})}{\partial \beta_{ij}} = (o_i\tau - \hat{o}_i)\frac{-y_ix_j}{1 - \beta_{ij}x_j} \tag{10}$$

*or $\partial L(\hat{\boldsymbol{o}}, \boldsymbol{o})/\partial \beta_{ij} = (o_i\tau - \hat{o}_i)\partial log(S(y_i, x_j; 1, \beta_{ij}))/\partial \beta_{ij}$ where $\hat{\boldsymbol{o}}$ is the target vector and $\boldsymbol{o}$ is the output vector and the **fully-connected synapse graph** outputs through a softmax activation function that is $o_j = softmax(y_j)$.*

*Proof.* The proof is given in the Appendix A. □

### 2.4.2 GRADIENT AND BOSE-EINSTEIN DISTRIBUTION

Considering the surprisal space, let $(u_k, v_k, \gamma_{ki}) = -log(x_k, y_k, \beta_{ki})$, the **fully-connected synapse graph** is denoted as $v_k = u_k + \sum_i(-log(1 - e^{-(\gamma_{ki}+u_i)}))$ . Compute the gradient over parameters

$$\frac{\partial v_k}{\partial \gamma_{pq}} = -\sum_i \frac{e^{-(\gamma_{ki}+u_i)}}{1 - e^{-(\gamma_{ki}+u_i)}}\frac{\partial \gamma_{ki}}{\partial \gamma_{pq}} = -\sum_i \frac{e^{-(\gamma_{ki}+u_i)}}{1 - e^{-(\gamma_{ki}+u_i)}}\delta_{kp}\delta_{iq} \tag{11}$$

because only when $k = p$ and $i = q$, two $\delta$ are 1, so $\frac{\partial v_p}{\partial \gamma_{pq}} = \frac{-e^{-(\gamma_{pq}+u_q)}}{1-e^{-(\gamma_{pq}+u_q)}}$. Replacing the indexes and reformulating, we have

$$\frac{\partial v_i}{\partial \gamma_{ij}} = \frac{-1}{e^{\gamma_{ij}+u_j} - 1} \tag{12}$$

The right side of Eq.(12) is the negative Bose-Einstein Distribution in the surprisal space.

To compute the loss function in surprisal space, we convert the target vector $\hat{\boldsymbol{o}}$ and output vector $\boldsymbol{o}$ to surprisal space as $(\hat{\boldsymbol{o}}, \boldsymbol{o})$, so the new loss function is $\mathcal{L}(\hat{\boldsymbol{t}}, \boldsymbol{t}) = \sum_k \hat{t}_k * t_k$. The $log$ function has been removed in $\mathcal{L}(\hat{\boldsymbol{t}}, \boldsymbol{t})$ because $log$ is implied in the surprisal space. Without using an activation function, there is $t_k = v_k$. By Eq.(12),

$$\frac{\partial \mathcal{L}(\hat{\boldsymbol{t}}, \boldsymbol{t})}{\partial \gamma_{ij}} = \sum_k \frac{\partial \mathcal{L}(\hat{\boldsymbol{t}}, \boldsymbol{t})}{\partial v_k} \frac{\partial v_k}{\partial \gamma_{ij}} = \frac{-\hat{t}_i}{e^{\gamma_{ij}+u_j} - 1} \tag{13}$$

We can apply error back-propagation to implement gradient descent for **synapse learning**.

$$\gamma_{ij} \leftarrow \gamma_{ij} - \eta \frac{\partial \mathcal{L}(\hat{\boldsymbol{t}}, \boldsymbol{t})}{\partial \gamma_{ij}} \tag{14}$$

where $\eta$ is the learning rate.

The equation Eq.(14) illustrates that the learning of synaptic neural network follows the Bose-Einstein statistics in the surprisal space. This paper "Memory as an equilibrium Bose gas" by (Fröhlich (1968), Pascual-Leone (1970)) showed that memory maybe possible to be represented as the equilibrium of Bose gas.

## 3 EXPERIMENTS

### 3.1 SYNAMLP : SYNAPTIC MULTIPLE LAYER PERCEPTRONS

We are going to illustrate a Synaptic Neural Network implementation SynaMLP with the connection of Multiple Layer Perceptrons (MLP) (Minsky et al. (2017)).

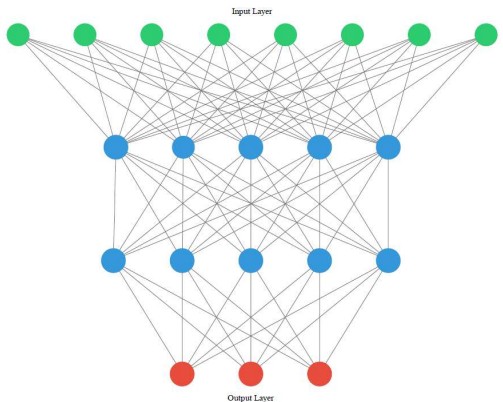

Figure 1: SynaMLP: (green, blue, red) dots are (input, hidden, output) layers.

SynaMLP has an input layer for down-sampling and an output layer for classification. The hidden layer is a block of fully-connected synapse tensor. The inputs of the block are neurons from the input layer and the outputs of the block are neurons to the output layer. The block is the implementation of synapse tensor in Eq.(9). The activation functions are connected **synapse tensor** to the output layer. Moreover, the input of the block is a probability distribution. The block can be thought of the replacement of the weight layer in a standard neural network.

## 3.2 MNIST Experiment

To proof-of-concept, we implemented the SynaMLP with MNIST. Hand-written digital MNIST data sets are used for training and testing in machine learning. It is split into three parts: 60,000 data points of training data (mnist.train), 10,000 points of test data (mnist.test), and 5,000 points of validation data (mnist.validation) (LeCun et al. (2010)).

| | Synapse MLP | Dense MLP |
|---|---|---|
| Iteration | 10001 | 10001 |
| Test loss | 0.0942904587357543 | 0.0906127168575471 |
| Test accuracy | **0.9802**000087499618 | **0.9830**000066757202 |

Table 1: SynaMLP MNIST Testing

The MNIST SynaMLP training and testing is implemented by Python, Keras and Tensorflow (Abadi et al. (2016)) from the revision of the example of mnist_mlp.py in Keras distribution. The synapse tensor is designed to be a class to replace Dense in Keras. The layer sequence is as below,

| Layer Type | Output Shape | Param # |
|---|---|---|
| Dense[1] | (None, 300) | 235500 |
| Batch-normalization[1] | (None, 300) | 1200 |
| Activation[1] | (None, 300) | 0 |
| **Synapse[1]** | (None, 300) | 90000 |
| Batch-normalization[2] | (None, 300) | 1200 |
| Dense[2] | (None, 10) | 3010 |
| Optimizer | Adam | default |

Table 2: SynaMLP Layers

In the comparison experiment, SynaNN MLP and traditional MLP generated the similar test accuracy of around 98%. We applied a softmax activation function in front of the input of synapse to avoid the error of NAN (computing value out of the domain). In fact, synaptic neural network handles a probability distribution (vector from neurons).

## 4 Conclusion

In this paper, we presented and analyzed a ***Synaptic Neural Network (SynaNN)***. We found the fine structure of synapse and the construction of synapse network as well as the BE distribution in the gradient descent learning. In surprisal space, the input of a neuron is the addition of the identity function and the sum of topologically conjugate functions of inhibitory synapses which is the sum of bits of information. The formula of surprisal synapse function is defined as

$$\mathcal{LS}(u, v; \theta, \gamma) = (\theta + u) + (\mathcal{I} \circ \mathcal{F} \circ \mathcal{I}^{-1})(\gamma + v)) \tag{15}$$

The non-linear synaptic neural network may be implemented by physical or chemical components. Instead of using a simple linear synapse function, more synapse functions maybe found in the researches and applications of neural network.

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

## APPENDIX A   GRADIENT EQUATION THEOREM

Let the loss function $L(\hat{o}, o)$ of the ***fully-connected synapse graph*** Eq.8 be equal to the sum of cross-entropy $L(\hat{o}, o) = -\sum_i \hat{o}_i log(o_i)$, then its item of parameter gradient is

$$\frac{\partial L(\hat{o}, o)}{\partial \beta_{ij}} = (o_i\tau - \hat{o}_i)\frac{-y_i x_j}{1 - \beta_{ij}x_j} \tag{16}$$

or

$$\frac{\partial L(\hat{o}, o)}{\partial \beta_{ij}} = (o_i\tau - \hat{o}_i)\frac{\partial log(S(y_i, x_j; 1, \beta_{ij}))}{\partial \beta_{ij}} \tag{17}$$

where $\hat{o}$ is the target vector and $o$ is the output vector and the ***fully-connected synapse graph*** outputs through a softmax activation function.

*Proof.* Given $o_j = softmax(y_j)$, then

$$\frac{\partial log(o_j)}{\partial y_k} = \delta_{jk} - o_k, \ \frac{\partial L(\hat{o}, o)}{\partial y_k} = o_k\tau - \hat{o_k} \tag{18}$$

where $\delta_{jk}$ is the Kronecker delta and $\tau = \sum_j \hat{o}_j$ is a constant. After applying log on both sides of Eq.8, we can compute the partial derivative over parameters as,

$$\frac{\partial log(y_k)}{\partial \beta_{pq}} = \sum_i \frac{-x_i}{1 - \beta_{ki}x_i}\frac{\partial \beta_{ki}}{\partial \beta_{pq}} = \sum_i \frac{-x_i}{1 - \beta_{ki}x_i}\delta_{kp}\delta_{iq} = \frac{-x_q}{1 - \beta_{kq}x_q}\delta_{kp} \tag{19}$$

then applying log derivative on the left side and multiple $y_k$ on both sides, we have

$$\frac{\partial y_k}{\partial \beta_{pq}} = \frac{-x_q y_k}{1 - \beta_{kq}x_q}\delta_{kp} \tag{20}$$

Since

$$\frac{\partial L(\hat{o}, o)}{\partial \beta_{pq}} = \sum_k \frac{\partial L(\hat{o}, o)}{\partial y_k}\frac{\partial y_k}{\partial \beta_{pq}} \tag{21}$$

from Eq.18 and Eq.20, we have

$$\frac{\partial L(\hat{o}, o)}{\partial \beta_{pq}} = \sum_k (o_k\tau - \hat{o_k})\frac{-x_q y_k}{1 - \beta_{kq}x_q}\delta_{kp} = (o_p\tau - \hat{o_p})\frac{-x_q y_p}{1 - \beta_{pq}x_q} \tag{22}$$

replace index $p, q$ by $i, j$ in Eq.22, we proved the gradient formula Eq.10 which is Eq.4 as well. $\square$

