# OpenReview forum: "A Synaptic Neural Network and Synapse Learning"
_ICLR.cc/2019/Conference_

### Official Review · AnonReviewer2 · 2018-10-31
**Complicated math, terminology and network to obtain standard MLP performance**

**Rating:** 2
**Confidence:** 3

**Review:**

Quality - poor
The highly complicated work is evaluated only on the simplest of benchmarks with no significant results.

Clarity - poor
The paper seems to amount to gobbledygook, many disparate terminology strung together.

Originality
No idea.

Significance
None.

cons: the paper to me seems a hashing of citations to the main works in neuroscience and deep learning for which only the simplest network is demonstrated (single hidden layer MLP on MNIST) with results that do not exceed that of a standard MLP.
pros: the only pro I can think of for this work is that synaptic computing imo deserves more consideration, as real synapses are very complicated beasts, the functioning of which relatively little is known about.

---

> ### Public Comment · (anonymous) · 2018-11-06
> **Authors deserve a polite review**
>
> I am shocked by the unnecessarily harsh tone of this review. I skimmed the paper and can see the problems the reviewers have with the paper, but I have no doubt that the authors have spent serious effort in this work.
>
> The tone of this review is hurtful and doesn't comply with good review practice.
>
> Please see AnonReviewer3, for how a negative review can be communicated in a neutral tone, and how feedback to authors should be given.

---

> ### Author Response · Authors · 2018-11-06
> **It is better to list evidences in details.**
>
> "the paper to me seems a hashing of citations to the main works in neuroscience and deep learning"
>
> Please show us what main works in neuroscience and deep learning we have been hashing?

---

### Official Review · AnonReviewer3 · 2018-11-03
**This paper is not presented with sufficient clarity for ICLR publication**

**Rating:** 2
**Confidence:** 3

**Review:**

The authors present a biologically-inspired neural network model based on the excitatory and inhibitory ion channels in the membranes of real cells.  Unfortunately, the paper is structured incoherently, making it nearly impossible to appreciate the authors' contribution.  The introduction references neuroscience alongside ResNets, FinTech, surprisal spaces, Bose-Einstein statistics, and topological conjugacy without adequately motivating or defining any of the above.  The fundamental definition of the model synapse as a conditional probability (Eq. 1) is not guaranteed to be non-negative, casting serious doubt on any of the subsequent conclusions.   Figure 1 conveys no further information about the proposed model.  There is no explicit related work or background section.  The single experiment offers no comparison to alternative methods.   I suggest the authors invest serious effort into rewriting the paper to clarify the presentation and explicitly state their contributions in the context of existing work on biologically-inspired learning models.  This is indeed a subfield of machine learning worthy of more investigation.

---

> ### Author Response · Authors · 2018-11-06
> **About the Clarity of this Paper**
>
> Thanks for your review.
>
> Q1: "Unfortunately, the paper is structured incoherently, making it nearly impossible to appreciate the authors' contribution."
>
> This paper was constructed in the structure of a math paper. We first defined a basic formula and gave an explanation of the formula that was based on an abstraction of the biological neural network. Then we gave the definition of the model, related concepts, and theorems with proofs. In this way, we explored and concluded many features of the model, the target was to figure out the learning rules be applied with backpropagation.  Finally, we showed an experiment to prove the concept.
>
> Authors' contributions: (A lot of)
>
> 1. The neural network model is the authors' creation.
>
> The biologically-inspired synapse equation S(x,y; a,b)=ax(1-by) is defined by the product of the opening probability of excitatory channels of a synapse and the opening probability of inhibitory channels of a synapse. Unlike the synapse in spike neural network, we consider the ion channels as the basis to build our synapse model. We ignored the spike feature of the biological neural network because it is the feature of neurons not synapses. The flow of ions and the random opening of the channels are the foundation of our synapse analysis. From a probability perspective, we abstract the synapse equation.  In contrast to classical neural network with weights, its synapse is simply a product of the input variable and the weight. Our synapse is a non-linear unit. The practical biological synapse is much complex, But we present a simple model that can be analyzed in math.
>
> 2. We defined the surprisal space to connect Information Theory with our model.
>
> Although entropy is widely used in machine learning but surprisal is the more fundamental concept. When we apply surprisal on synapse equation, we have a linear combination in log space which has been used in machine learning analysis. With a negative in front of the log function, we can convert data to the surprisal space. In surprisal space, we can explain the negative log probability as the bits of self-information. That does make sense of surprisal space. It can be a new representation of the neural network. If somebody finds papers to applying surprisal space to explain neural network please let us know, we are going to list them as our references.
>
> By defining surprisal space, we build the mapping between a probability space and the surprisal space. It is a real positive space in our definition. The bits addition is the basic operation of a neural network.
>
> 3. Synapse with topological conjugacy is our discovery
>
> It is very exciting to find that the surprisal of the inhibitory probability is a topological conjugation in our model. That means the dynamical behavior in probability space can be bijected into surprisal space and both have the same dynamics.
>
> In advance, we discovered that this topological conjugation is a commutative diagram in category theory. That opened the door to apply new mathematic tools to study neural network. The discovery between the connection of the neural network and category theory is unexpected. That is one of our exciting contributions.
>
> 4. We discovered gradient updating in synapse learning followed Bose-Einstein distribution
>
> This is a direct conclusion from synapse equation in surprisal space without any statistical hypothesis.  That solved the famous black box problem in our model. So we can expect some kind of BE distribution in the parametric matrix. It is a new representation of a neural network.
>
> 5. We constructed a fully-connected synaptic neural network as synapse tensor
>
> We successfully convert fully-connected non-linear synapse network into a matrix (tensor) computing. This synapse tensor is a special connection of synapses. Other network topologies are possible. Synapse tensor can be basic blocks to construct a large-scale neural network.
>
> 6. We discovered that synaptic neural network has a similar block to ResNet block in surprisal space
>
> That is why we mentioned ResNet.  Except we apply surprisal non-linear function but still computing identity mapping.
> So we expect some features of ResNet such as protect gradient from vanishing in the very deep synaptic neural network.
>
> 7. We proved the gradient rule with loss function that mapped in surprisal space
>
> That is proof that we can apply the backpropagation algorithm on the fully-connected synaptic neural network.  The proof is in very details because we want to verify that the new gradient computing is correct.
>
> In conclusion, our synaptic neural network is compatible with backpropagation, however, spike neural network is not.

---

> > ### Author Response · Authors · 2018-11-06
> > **About the Clarity of this Paper (continued ...)**
> >
> >  Q2: "The introduction references neuroscience alongside ResNets, FinTech, surprisal spaces, Bose-Einstein statistics, and topological conjugacy without adequately motivating or defining any of the above."
> >
> > From the answer of Q1, we expect reviewers to know how these things are connected together.
> >
> > For FinTech mentioned, it is an application and we are going to remove it because we have removed one of our swap equation. So readers may think it from nowhere.
> >
> >
> >
> > Q3: "The fundamental definition of the model synapse as a conditional probability (Eq. 1) is not guaranteed to be non-negative, casting serious doubt on any of the subsequent conclusions."
> >
> > You are right. At first, we have considered limiting the value of alpha and beta to be less than 1.0. But we want to know more the range of their values in practice. In some cases, alpha can be bigger than 1.0 and make sense. But beta looks always less than 1.0. Yes. we are going to limit beta less than or equal to 1.0. And in surprisal space, beta is always less than 1.0.
> >
> > Q3: "Figure 1 conveys no further information about the proposed model."
> > Figure 1 is to help readers to understand the probability explanation of the synapse equation Eq. 1.
> >
> > Q4: "There is no explicit related work or background section. "
> > The paper is in full 8 pages, remove some contents to add background section? The related works are linked in the references.

---

### Official Review · AnonReviewer1 · 2018-11-07
**Not clear if this is -- or will be -- a practically useful approach**

**Rating:** 3
**Confidence:** 3

**Review:**

The authors propose a hybrid neural network, composed of a synapse graph that can be embedded into and a standard neural network, such that the entire architecture can be trained in a way that is compatible with the gradient descent and backpropagation of. As a proof of concept, the hybrid architecture is trained to classify MNIST.

I am not convinced by the way this work is motivated. What problem are the authors actually addressing? Just because biological neurons use synapses does not mean we should try hard to put a certain instance of them into deep neural networks. Clearly this is not an attempt to add to neuroscience, as beyond the inspiration of neurons having synapses, there is little attempt to biologically plausible. As an attempt to add to machine learning research, the neuroscience motivation is unconvincing. Provided that the math works out (and I admit that I did not attempt to follow the detailed derivations), this looks like an interesting intellectual exercise, but it also seems a bit like a discovery of a hammer that is in need for nails to be applied to. And it’s not even clear to me how practical the hammer would actually be, even if we had a convincing problem setting at hand. How scalable is it beyond toy-settings? The final sentence makes a tantalizing claim, but at this stage the work has to resort to promising potential, rather than being able to demonstrate that it is practically useful.

Moreover, this work is not presented right for the venue and audience, and would need substantial rewriting and restructuring to make the central claims and contributions sufficiently clear.

---

> ### Author Response · Authors · 2018-11-07
> **What doors we have opened (1)**
>
> Recall the history of the neural network, current neural networks come from the simulation of biological neurons and their systems. An artificial neural network is the simplified mathematical model of the biological neural network. The deep neural network is far more simple in topology than the human brain. History has proved the method to study simplified neuron model and expand it to bring rich fruits. In reverse, we know more brain from model studying.
>
> The motivation of this work is based on the research and analysis of the biological neural system.  First, synapse plays an important role in learning and memory. In neuroscience, it is synaptic plasticity. The excitation and inhibition are observed in the synapse network. In advanced, the random opening of ion channels is also observed. Second, synapses can connect to other synapses to form a synapse network. Third, synapse makes a non-linear transform.
>
> Synapse in the artificial neural network is supposed as a simple linear amplifier, it is the multiplication of a parameter and an input variable.  There are no synapses connecting to synapses. But one thing is the same, the learning and memory are related to synapses, the change of synaptic parameters. Current artificial neural network ignored the existence of synapses but simply consider them as weights. All the focus are on neurons.
>
> Back to our motivation. Why synapse act as a transform?  Why is it non-linear? What is the distribution of the synaptic matrix? The same question for an artificial neural network is what is the distribution of its weight matrix? Make sense?
>
> OK. We found a reasonable synapse function. The reasons were explained in the paper.  The function is not a simulation of a synapse but an abstraction of the probability. It is a non-linear function with parameters.  From their connection, we can form a synapse network.
>
> What is our "interesting intellectual exercise" bring in our paper?
>
> 1. Probability Space and Surprisal Space
>
> Logarithmic space has been studied in artificial neural network for a long history. But fruitless in ANN. Because the data field of ANN is the real number field from negative to positive. The surprisal has been defined in information theory and natural language processing and it is related to a random variable. Direct select variable from probability space,  the surprisal is the negative log function. Moreover, we found that it is useful to define the surprisal space. Two difference between logarithmic space and surprisal space: 1) different in a negative sign 2) real space vs (0,1) space.
> surprisal represents a self-information bit. The non-linear product in probability space is the linear addition in surprisal space. There may have a lot of new things need to be studied in the surprisal space. Surprisal Space opens a door.

---

> > ### Author Response · Authors · 2018-11-07
> > **What doors we have opened (2)**
> >
> > In our paper, the order of the description is in reverse order in history.
> >
> > 2. Topological Conjugation
> >
> > Our synapse function worked in probability space. How about it worked in surprisal space? It is a linear function plus a log function. In this log form, there is no universal meaning. By chance, we figured out the item with log function is a kind of topologically conjugated function. That is a map between probability space and surprisal space.
> >
> > This result is exciting. Because we know synapse is doing a type of topological transform, at least in our model. Any help for machine learning or AI today, we do not know. But it opens a door to design new synapse function.
> > Hope it is helpful in learning and memory research.
> >
> > 3. Gradient in Bose-Einstein Distribution
> >
> > This result is achieved by computing derivation over parameters. In our model, the update of parameters follows BE distribution. So we know something about the black box of the parametric matrix.
> >
> > 4. Learning by Backpropagation
> >
> > We successfully represented the fully-connected synaptic neural network in Tensorflow/Keras. So it can be a block to construct any scale neural network. It can also be used with other blocks to form a hybrid neural network.
> >
> > Since we have the explicit expression of the Jacobian matrix of synapses, we can study new gradient algorithms.
> >
> > 5. How scalable is it beyond toy-settings?
> >
> > In Synaptic Neural Network, we have shown that a synapse is like a ResNet block in surprisal space. That is
> > G(u) = u + F(v).  It has the feature to avoid gradient vanishing. So deep layers can be applied to avoid error increasing.

---

### Official Review · AnonReviewer4 · 2018-11-08
**Not understandable**

**Rating:** 2
**Confidence:** 4

**Review:**

Thanks for submitting your paper. It takes a lot of effort and courage to put your ideas out into the world. Sometimes the hardest work for researchers is conveying their thoughts to others in a manner in which those ideas can be understood.

With that in mind, I had an extremely difficult time following your arguments.

I noticed several things:
 - There are numerous places in the text that lack proper citation, or are cited improperly.

 - Why was there not a related methods section? I find it hard to believe that all of your ideas have no precursor.

 - When there are citations, there is usually only one text and it is quite old. For example, all of your neuroscience citations reference a work that is almost 40 years old. There have been quite a few improvements in our biological understanding as well as theoretical understanding since then. ( I make this point as a common justification used in the manuscript is that the method describes how synapses function in biology. )

- There is a claim regarding how this can be used in fintech. This statement doesn't belong in this work.

 - There are many different equations given throughout the text. Some of these equations come from areas like physics or information theory, and others seem to be of your own design. Regarding the latter, there is no justification or explanation for the origin of the equations. Regarding the former, if you are using equations from lots of different fields, or even field you think part of your audience might not be familiar with, you should, at the very least, include a some description of the algorithm or intuition as to why it is being leveraged.

 - It wasn't clear from your diagrams or your descriptions what the difference between a synapse and a neuron was in your architecture. It seemed like the name was used interchangeably in some areas, but then had a strict definition in others.

 - I was also not able to understand how the excitatory and the inhibitory connections that were to enter each neuron were connected to the previous layer of the network. Is a link between neurons in Figure 2 actually two links? If this is the case, then it is a direct violation of Dale's law. Again, I only mention this because most arguments seem to be of the form "this is correct because it is how it is done biologically".

 - There were a few claims made in the paper that were completely unsubstantiated. A good example of this was in the conclusion section part ii) where it was stated that "using a large number of synapses and neurons SynaNN can solve the complex problems in the real world."

- Also, the last sentence of the conclusion was not discussed anywhere in the rest of the paper. Nor was the statement itself supported except with a single citation and no description.

Regarding the empirical testing of your algorithm, I was very dissapointed to see that the only dataset it was tested against was MNIST. Furthermore there was absolutely no benchmarking against other comparative algorithms. At the very least I would have expected a comparison to the perceptron algorithm that you use as inspiration, but that would also still not have been enough.

This paper needs heavy amounts of work to make it understandable. Once it is understandable an attempt to evaluate the merit of the scientific contribution would then be possible.

---

> ### Author Response · Authors · 2018-11-09
> **Answer to reviewer (1)**
>
> Thanks for your detail review.
>
> With that in mind, I had an extremely difficult time following your arguments.
>
> I noticed several things:
>  - There are numerous places in the text that lack proper citation, or are cited improperly.
>
> We are going to check it. We are in the procedure to reorganize and revise the paper.
>
>  - Why was there not a related methods section? I find it hard to believe that all of your ideas have no precursor.
>
> We are going to explain some of our methods. For example, dual space analysis is a widely applied method. In our case, it is probability space and surprisal space. It is hard to study the non-linear function in the probability space,
> but our function becomes linear in surprisal space. Many results can be easily concluded from surprisal space. Entropy, surprisal, surprisal function have been well known and defined. But we need to consider everything of synapse after converting to surprisal, therefore we defined the surprisal space. We do not know somebody defined and use it so far. Let us know if find some references.
>
> Log space has been widely used. Surprisal space different from log space in a negative sign and the domain. The domain of log space is the field of the positive real number. The domain of surprisal space is the real interval of (0,1). Surprisal space does make sense as the self-information bit space.
>
>  - When there are citations, there is usually only one text and it is quite old. For example, all of your neuroscience citations reference a work that is almost 40 years old. There have been quite a few improvements in our biological understanding as well as theoretical understanding since then.  ( I make this point as a common justification used in the manuscript is that the method describes how synapses function in biology. )
>
> You are right. We have no intention to ignore the latest remark work in neuroscience. Most synaptic models are built on the differential equations of electronic property. Our biological model is very simple but we believe that it works in the right direction. The opening property of ion channels with different types is the key in our analysis. Fortunately, there are two types of channels in a synapse: excitatory and inhibitory channels. The randomness of the channels is the basis for us to apply probability theory. It is very difficult to figure out a complete synapse model. We tried to build a simple model to approach the property of the synapse.
>
> - There is a claim regarding how this can be used in fintech. This statement doesn't belong in this work.
>
> This paragraph is wrong. It will be removed.
>
>  - There are many different equations given throughout the text. Some of these equations come from areas like physics or information theory, and others seem to be of your own design. Regarding the latter, there is no justification or explanation for the origin of the equations. Regarding the former, if you are using equations from lots of different fields, or even field you think part of your audience might not be familiar with, you should, at the very least, include some description of the algorithm or intuition as to why it is being leveraged.
>
> One confusion is that we put some standard terms such as entropy in our definition. We are going to make all exact definitions and theorems from us.
>
> One case is to prove the gradient of the synapse function in surprisal space has the expression Bose-Einstein distribution. That is dlog(1-e^-x)/dx = 1/(e^x-1). Unfortunately, we did not found any references to mentation this. All of them came from the computing in statistical physics. In our context, it has an obvious meaning that is the gradient over the parameter of our synapse function in surprisal space. As far as we know this is the first time we figured out this equation, at least a rediscovery. We'd like to know any claims for their discovery.
>
> Yes, we are going to add some explanations.

---

> ### Author Response · Authors · 2018-11-09
> **Answer to reviewer (2)**
>
>  - It wasn't clear from your diagrams or your descriptions what the difference between a synapse and a neuron was in your architecture.  It seemed like the name was used interchangeably in some areas but then had a strict definition in others.
>
> You can think of synapse graph as the weight layer in the classical artificial neural network. We are going to find clarity.
>
>  - I was also not able to understand how the excitatory and the inhibitory connections that were to enter each neuron were connected to the previous layer of the network. Is a link between neurons in Figure 2 actually two links? If this is the case, then it is a direct violation of Dale's law. Again, I only mention this because most arguments seem to be of the form "this is correct because it is how it is done biologically".
>
> Figure 2 actually represents a synapse tensor or matrix. It has n inputs and n outputs with n*n synapses.
>
> You raised a very interesting question. We defined a synapse with two inputs and one output. Ignored the parameters it is x(1-y). It is hard to figure out how it connects to a neuron in this probability space. After convert to surprisal space, it is -(log(x)+log(1-y) or I(x)+G(y). For multiple synapses contacted to one neuron, it is I(x)+Sum(G_i(y_i)). That follows Dale's law. Since I(x) is the Identity function, G(y) is the topologically conjugated function, we may think a synapse is to do a topologically conjugated transform in surprisal space.
>
> In our wild conjecture, the synapse may change its "shape" to complete its function. At present we do not know what the "shape" is.
>
>  - There were a few claims made in the paper that was completely unsubstantiated. A good example of this was in the conclusion section part ii) where it was stated that "using a large number of synapses and neurons SynaNN can solve the complex problems in the real world."
>
> This claim is based on the function block comparison between SynaNN and ResNet. Both can solve the gradient degradation problem. So the error will not increase as the layers increase. Yes, we can remove this claim. It is not a reality today but we believe that it is true in the future.
>
> - Also, the last sentence of the conclusion was not discussed anywhere in the rest of the paper. Nor was the statement itself supported except with a single citation and no description.
>
> There is somebody to build a quantum neural network from bosonic sampling. BEC is possible to do quantum computing.
>
> - Regarding the empirical testing of your algorithm, I was very disappointed to see that the only dataset it was tested against was MNIST. Furthermore, there was absolutely no benchmarking against other comparative algorithms.
> At the very least I would have expected a comparison to the perceptron algorithm that you use as inspiration, but that would also still not have been enough.
>
> We are going to do more experiments. For MLP the best result is 98.4% in less than 5 layers with a classical artificial neural network. Our result is not bad. For any claims that can easily be over 99%, they are not MLP. MLP is only a proof of concept.

---

> ### Author Response · Authors · 2018-11-09
> **Comparison Testing**
>
> Thanks to your suggestion to do a comparison testing.
>
> We have done two MLP tests in the same configuration of Keras/Tensorflow python code.
> The only difference is to replace Dense layer by Synapse layer in the hidden layer.
> Both the input layer and output layer are Dense.
>
> Keras/Synapse:
> Test loss: 0.09429045873575433
> Test accuracy: 0.9802000087499618
>
> Keras/Dense:
> Test loss: 0.09061271685754718
> Test accuracy: 0.9830000066757202
>
> The accuracy is not such a disappointment. Everybody, including us, has thought MLP can achieve 99% in any way. In our testing, without BatchNormlization, it is even hard to achieve 98%. This reminds us many disappoint results on MNIST from other models such as Spike Neural Network. The intrinsic limitation of MLP may be the reason for the poor results. The model itself is not wrong.
>
> Below is the configuration.
>
> Using TensorFlow backend.
> 60000 train samples
> 10000 test samples
> _________________________________________________________________
> Layer (type)                 Output Shape              Param #
> =================================================
> dense_1 (Dense)              (None, 300)             235500
> _________________________________________________________________
> batch_normalization_1 (Batch (None, 300)   1200
> _________________________________________________________________
> activation_1 (Activation)    (None, 300)           0
> _________________________________________________________________
> synapse_1 (Synapse)          (None, 300)          90000
> _________________________________________________________________
> batch_normalization_2 (Batch (None, 300)   1200
> _________________________________________________________________
> dense_2 (Dense)              (None, 10)                3010
> =================================================
> Total params: 330,910
> Trainable params: 329,710
> Non-trainable params: 1,200
> _________________________________________________________________
> Train on 60000 samples, validate on 10000 samples
> Epoch 1/30

---

> > ### Author Response · Authors · 2018-11-09
> > **Here is the all Dense configuration**
> >
> > Using TensorFlow backend.
> > 60000 train samples
> > 10000 test samples
> > _________________________________________________________________
> > Layer (type)                 Output Shape              Param #
> > =================================================================
> > dense_1 (Dense)              (None, 300)               235500
> > _________________________________________________________________
> > batch_normalization_1 (Batch (None, 300)               1200
> > _________________________________________________________________
> > activation_1 (Activation)    (None, 300)               0
> > _________________________________________________________________
> > dense_2 (Dense)              (None, 300)               90300
> > _________________________________________________________________
> > batch_normalization_2 (Batch (None, 300)               1200
> > _________________________________________________________________
> > dense_3 (Dense)              (None, 10)                3010
> > =================================================================
> > Total params: 331,210
> > Trainable params: 330,010
> > Non-trainable params: 1,200
> > _________________________________________________________________
> > Train on 60000 samples, validate on 10000 samples
> > Epoch 1/30

---

### Author Response · Authors · 2018-11-19
**Revision paper has been submitted.**

Followed the comments and suggestions of reviewers, we have made a large amount of revision on the paper. The new PDF file is available on OpenReview now.

1. We have revised the organization of the paper to increase the readability so that the logic of paper is clarity.

2. A background section has been added to explain our motivations, problems raised and solutions.

3. More references are added to explain related works. Many references do not directly contribute to the problem solving,  however,  after the solution, we find the related works. During inference, we may need only fundamental scientific and technological knowledge and skills.

4. We try to present the concepts and methods as easy to understand as possible.  In spite of some modern mathematical concepts and methods are applied, you can understand them without pre-required knowledge.

5. We list contributions in a paragraph. Four theorems describe our key contributions. While most of the proofs are short, a longer proof is in appendix A.

Appreciated reviewers' valuable comments and reading.

---

> ### Comment · AnonReviewer4 · 2018-11-26
> **Better, but unfortunately still needs lots of work**
>
> After reading your new submission I can see you spent lots of time on revisions and tried to address many my questions as well as those of the other reviewers. Thank you for your effort in those regards. Unfortunately, there are still major issues with the readability of the paper from a language usage standpoint as well as a conceptual standpoint. I will address the conceptual points that jumped out to me while reading this revision.
>
> 1) There seems to be some confusion on your part regarding the nature of synapses and how they behave. You are correct that there are channels in the synapse that allow different ions to flow across them. The voltage difference across the cell membrane causes influx and efflux of ions. These ions flow through channels that are able to filter out specific ions based on the size of the ion and the folding abilities of the proteins that make up the ion channel. You are also correct that the opening and closing of specific channels is stochastic. However, the population activity of these channels is not entirely stochastic at the synapse level. These channels are not themselves inhibitory or excitatory but rather the entire synapse is either excitatory or inhibitory. Furthermore, you go on to say that it is known in neuroscience that most synapses are inhibitory. I do not believe that this is an accurate statement for the brain as a whole, or for even most sub areas of the brain. To be honest, I'm not sure anyone as yet answered this question as it is very complicated to do this measuring and mapping.
>
> 2) It seems that one of the things that could have greatly enhanced the quality of your paper, i.e. experiments on data more difficult than MNIST, was not included even though a benchmark against a standard MLP was shown.
>
> 3) The definition of a synapse seems very trivial and there is no discussion as to where it comes from. You cite an unpublished paper as the source, but that is all. Since the current manuscript fundamentally relies on it, it creates a problem if it isn't well described.
>
> 4) Although I was able to follow your arguments better in this revision, the novelty of the discoveries does not seem very high. If I understand correctly, your synapse network is essentially a more complicated version of dropout applied to your graph edges because only when excitation(x) is high and inhibition (y) is low does your function S return high values. And since the population of inhibitory should be equal to or smaller than the excitatory, a high value of y will always return low values of the function S.

---

> > ### Author Response · Authors · 2018-11-26
> > **Response to AnonReviewer4 (1)**
> >
> > "After reading your new submission I can see you spent lots of time on revisions and tried to address many my questions as well as those of the other reviewers. Thank you for your effort in those regards. Unfortunately, there are still major issues with the readability of the paper from a language usage standpoint as well as a conceptual standpoint. I will address the conceptual points that jumped out to me while reading this revision."
> >
> > Thanks for your detail comments.
> >
> > "1) There seems to be some confusion on your part regarding the nature of synapses and how they behave. You are correct that there are channels in the synapse that allow different ions to flow across them. The voltage difference across the cell membrane causes influx and efflux of ions. These ions flow through channels that are able to filter out specific ions based on the size of the ion and the folding abilities of the proteins that make up the ion channel. You are also correct that the opening and closing of specific channels is stochastic."
> > OK, we agree on these observations of biological synapses.
> >
> > "However, the population activity of these channels is not entirely stochastic at the synapse level. These channels are not themselves inhibitory or excitatory but rather the entire synapse is either excitatory or inhibitory."
> >
> > Our hyperthesis is that the opening of some channels will enhance the activation of the neuron and the opening of some channels will inhibit the activation of the neuron. They are two classes: maybe called alpha-channel and beta-channel. (To avoid confusion, it may not call them excitatory channel and inhibitory channel). Here we just need to distinguish the contribution of channel's openness.
> >
> > The excitatory or inhibitory of  a synapse is decided by the synapse function. The excitatory and inhibitory are relative terms. The more alpha-channel opened the more contribution of a synapse to active the neuron; however, the more beta-channel opened the less contribution of a synapse to active the neuron.
> >
> > "Furthermore, you go on to say that it is known in neuroscience that most synapses are inhibitory. I do not believe that this is an accurate statement for the brain as a whole, or for even most sub areas of the brain. To be honest, I'm not sure anyone as yet answered this question as it is very complicated to do this measuring and mapping."
> >
> > This was an observation that many inhibitory synapses played more during activity. We are not sure whether this is right or not. But in our digital topological link of the network, we found that inhibitory item may connect more. For example, we may have one excitatory item x and all others are inhibitory y. It could be that all excitatory items can be packed into one item.
> >
> > "2) It seems that one of the things that could have greatly enhanced the quality of your paper, i.e. experiments on data more difficult than MNIST, was not included even though a benchmark against a standard MLP was shown. "
> >
> > That is our work in progress. We have constructed a new block to replace LSTM block in RNN. But have not found a right benchmark by using RNN. The CNN is the next project. The hard part is to figure out its tensor representation.
> >
> > MNIST is for proof-of-concept. We think this experiment has proved our model worked in practice.
> >
> > "3) The definition of a synapse seems very trivial and there is no discussion as to where it comes from. You cite an unpublished paper as the source, but that is all. Since the current manuscript fundamentally relies on it, it creates a problem if it isn't well described."
> >
> > It comes from our abstract biological model and probability computing we have discussed in the paper. It is the joint probability of two events: the opening probability of excitatory channels and the opening probability of inhibitory channels.
> >
> > The synapse function is not trivial. Its dynamics can be very complicated. Let's see S=4*x*(1-x). It is the case of y=x. It is the logistic function with chaotic dynamics. Iteration x <- 4*x*(1-x) in computer can watch chaos effects.
> > In our paper, we have shown the complicated structure from this simple and basic function.
> > We have discovered that S = x(1-y) is the addition of an Identity function and a topological conjugate function in surprisal space. Furthermore, this proved the deep meaning of the "trivial" synapse function.

---

> > > ### Author Response · Authors · 2018-11-26
> > > **Response to AnonReviewer4 (2)**
> > >
> > > "4) Although I was able to follow your arguments better in this revision, the novelty of the discoveries does not seem very high. "
> > >
> > > The discovery of the synapse function as the addition of an identity function and a topological conjugate function only is a high novelty.
> > >
> > > "If I understand correctly, your synapse network is essentially a more complicated version of dropout applied to your graph edges because only when excitation(x) is high and inhibition (y) is low does your function S return high values. And since the population of inhibitory should be equal to or smaller than the excitatory, a high value of y will always return low values of the function S."
> > >
> > > It is totally different from the dropout network. It is a bit similar to a Bayesian Network. From fundamental and real implementation synaptic neural network is totally different from the dropout. Synaptic Neural Network is a model, Dropout is an algorithm or strategy. Both are not comparable.
> > > Given an example,
> > >
> > > y1 = x1*(1-b1*x1)*(1-b2*x2) = x1*(1-(b1*x1+b2*x2)+b1*b2*x1*x2)
> > >
> > > this is an output of two synapses in the synapse tensor connection. It includes a self-correlated non-linear. The right side includes a linear item and a non-linear item x1*x2. We do not know how it is represented by dropout from algorithm viewpoint. For more variables, the non-linear effects are high degree polynomials. We do not see how dropout handles this.
> > >
> > > We can apply dropout in the output of the synapse network. Many outputs near zeros can be dropped out. In our testing example, we actually tested the dropout algorithm. We also applied Batch Normalization to generate better results in our MLP testing.
> > >
> > > "And since the population of inhibitory should be equal to or smaller than the excitatory, a high value of y will always return low values of the function S."
> > >
> > > We do not think this claim is correct. S is a non-linear function. No conditions suppose that inhibitory should be equal to or smaller than excitatory. These two can be independent events. both x and y can be from 0 to 1. And S=x*(1-y) can also be from 0 to 1.

---

> ### Comment · AnonReviewer3 · 2018-11-26
> **Thank you for your revisions**
>
> Thank you for the many revisions to the paper and the detailed responses to the authors.  Please continue to refine your ideas and presentation based on this feedback.  While the revisions are step in the right direction, the paper still needs to do a better job of communicating the problem that is being solved, the novel insights and key contributions, and the evidence for and against the proposed approach.  I concur with the specific suggestions from AnonReviewer4 below.

---

> > ### Author Response · Authors · 2018-11-26
> > **Response to AnonReviewer3**
> >
> > Thanks for your comments and suggestions.

---

### Author Response · Authors · 2018-12-03
**In 6 Layers CNN We Achieved 86% Accuracy of CIFAR10**

Replacing Fully Connected (FC) layer by SynaMLP (Synaptic Neural Network for Multiple Layer Perceptrons) in a 6 layer CNN neural network, we have achieved 86% accuracy CIFAR10. That is near equal to the classical neural network in the same settings.

Considering that the fully connected layer is to leaning a nonlinear function for classification, the synaptic neural network is suitable for this purpose. This experiment verified the capability of the synaptic neural network again.

---

### Meta-Review · Area_Chair1 · 2018-12-12
**early work with possible merit**

**Confidence:** 5
**Recommendation:** Reject

**Metareview:**

In this paper, neural networks are taken a step further by increasing their biological likeliness.  In particular, a model of the membranes of biological cells are used computationally to train a neural network.  The results are validated on MNIST.

The paper argumentation is not easy to follow, and all reviewers agree that the text needs to be improved.  ˜The neuroscience sources that the models are based on are possibly outdated.  Finally, the results are too meagre and, in the end, not well compared with competing approaches.

All in all, the merit of this approach is not fully demonstrated, and further work seems to be needed to clarify this.